# Social distancing practice and associated factors in response to COVID-19 pandemic at West Guji Zone, Southern Ethiopia, 2021: A community based cross-sectional study

**Anteneh Fikrie**⬛ *, **Elias Amaje, Wako Golicha**

School of Public Health, Institute of Health, Bule Hora University, Bule Hora, Ethiopia

* antenehfikrie3@gmail.com

## Abstract

### Background

Curtailing physical contact between individuals reduces transmission and spread of the disease. Social distancing is an accepted and effective strategy to delay the disease spread and reduce the magnitude of outbreaks of pandemic COVID-19. However, no study quantified social distancing practice and associated factors in the current study area. Therefore, the study aimed to assess social distancing practice and associated factors in response to COVID-19 pandemic in West Guji Zone, Southern Ethiopia, 2020.

### Methods and materials

A Community based cross-sectional study design was conducted among randomly selected 410 household members of Bule Hora Town, West Guji Zone. Data were collected by pretested interviewer administered structured questionnaire adapted from previous peer reviewed articles. The data were coded and entered in to Epi data version 3.5 and analyzed by SPSS version 23. The bivariate and multivariate logistic regressions analysis was done to identify factors associated with social distancing practice. Adjusted odds ratio with 95% confidence interval and p value <0.05 were used to declare statistical significance.

### Result

Out of 447 planned samples, 410 participants were successfully interviewed and included into final analysis; making the response rate of 91.7%. The median (±IQR) age of study participants was 28(±9) years. In this study, 38.3% [95% CI: 33.5%, 43.1%)] of the study participants have good social distancing practices for the prevention of COVID-19. Age group 26–30 years [AOR = 2.56(95% CI: 1.18–5.54)] and 31–35 years [AOR = 3.57(95%CI: 1.56–8.18)], employed [AOR = 6.10(95%CI: 3.46–10.74)],poor knowledge [AOR = 0.59 (95% CI:0.36–0.95)], negative attitude [AOR = 0.55 (95% CI:0.31–0.95)] and low perceived susceptibility [AOR = 0.33(95%CI: 0.20–0.54)] were significantly associated with good social distancing practice.

**Data Availability Statement:** All relevant data are within the manuscript and its Supporting Information files.

**Funding:** Bule Hora University has supported the study financially. The funder had no role in study design, data collection and analysis, decision to publish, or preparation of the manuscript.

**Competing interests:** The authors have declared that no competing interests exist.

## Conclusion

Social distancing practice is relatively poor in the study area. The knowledge and attitude level of participants were identified to be the major factors for the observed poor social distancing practice. Sustained efforts to improve awareness and attitudes towards COVID-19 prevention might improve adherence to social distancing practices.

## Introduction

Corona virus Disease 2019 (COVID -19) caused by the novel coronavirus (SARS CoV-2) has posed a public health emergency and a global crisis rapidly as of December 2019 originated in Wuhan, a city in the Hubei Province of China [1]. The viruses are a large family of viruses that cause illnesses ranging from common cold to more severe diseases such as Middle East respiratory syndrome (MERS-CoV) and severe acute respiratory syndrome (SARS-CoV-1). SARS-CoV -2 is a novel coronavirus that has not been previously identified in humans [2]. As of now, the source of the outbreak is unknown with certainty. However, it is believed that the virus might have link with a wet market (i.e. seafood's and live animals) from the Wuhan city [2,3]. The important mode of virus transmission is via person-to-person occurring mainly via respiratory droplets, and by contact with contaminated surfaces [1,4]. According to the World Health Organization (WHO), symptoms of infection with the virus include fever, cough, and shortness of breath and breathing difficulties. Severe infection can lead to pneumonia, multiple organ failure and even death [5].

The World Health Organization (WHO) has declared COVID-19 as a pandemic on 12th March 2020 [6]. As of 1 December, more than 254 million cases and 5.1 million deaths have been reported globally until 15 November 2021. The highest burden of the disease is in WHO American region with more than 95, 120, 017 (37.4%) confirmed cases recorded so far, whereas the lowest record 6, 186, 377 (2.4%) and 9, 794, 363(3.5%) of cases reported from WHO African and Western Pacific region respectively [7]. In Africa, the numbers of COVID-19 cases and impacted countries have been increasing steadily and there are no virus free countries in the region. Thus, South Africa, Kenya, and Ethiopia report higher number of new cases [8]. In Ethiopia 368,822 cases and 6,623 deaths were reported until 15 November 2021 [7]. However, reported statistics is likely to represent an underestimation of the true burden of the disease owing to shortcomings in active surveillance and diagnostic capacity of the country [9]. Across the globe, countries have been implementing different disease control and prevention measures to combat the pandemic with the objective of slowing disease transmission and reducing associated morbidity and mortality [6]. These measures include case identification, testing, isolation and care for all cases, tracing and quarantine of all contacts, social distancing at individual and community levels [10]. Moreover, SARS-CoV-2 has affected several countries across the globe, prompting governments to impose social distancing measures to slow the spread of infection. Ethiopia is also implementing the social distancing measures to reduce the spread of the virus [11].

Social distancing is one category of non-pharmaceutical interventions (NPI) which means making changes in our everyday routines in order to minimize close contact with others, including: avoiding crowded places and non-essential gatherings, avoiding common greetings, such as handshakes, limiting contact with people at higher risk (e.g. older adults and those in poor health), keeping a distance of approximately 2 meters from others to reduce risk of infection [12,13]. Staying at least six feet (i.e. 2 meter) away from other people reduces risk of

acquiring COVID-19 [14]. Evidence from past influenza pandemics revealed that social distancing practice reduces spread of the virus [15,16]. A study aimed at identifying whether controlling epidemic spread by social distancing do it well or not at all concluded that social distancing is the most cost-effective strategy of controlling the epidemic [17]. Particularly, during the early phase of the pandemic where neither proven treatment nor vaccination is available, implementation of non-pharmaceutical interventions (NPIs) like social distancing is an effective and option [18].

Social distancing changes the behavior of an individual that prevent disease transmission by reducing contact rates, but the benefits depend on the extent to which it is practiced by individuals. In the absence of other intervention measures, optimal social distancing reduces the risk by 30% [19]. On the other hand, according to European Centre for Disease Prevention and Control technical report it was estimated that if social distancing had been conducted one week, two weeks, or three weeks earlier in China, the number of COVID-19 cases could have been halted by 66%, 86%, and 95%, respectively [12]. A study conducted among government employees in Ethiopia revealed that, more than nine-in-ten (94.8%) avoids handshaking; whereas 89.5% and 88.1% practiced physical distancing and avoided mass gatherings and crowded places respectively [20]. Another study conducted in the same area found that the majority of respondents had reflected good knowledge, positive attitude and low magnitude of practice regarding COVID-19 prevention activities [21]. A nationwide online cross-sectional survey conducted in Uganda found that 14.7% of participants were not practicing social distancing [22].

The study was conducted after the government underwent series of lockdown as a major means of limiting the spread disease. So this would reveal whether the communities are adhered or not to COVID-19 prevention measures particularly of the social distancing practices. More importantly, in resource constrained countries like Ethiopia, social distancing is an effective and affordable way of containing the pandemic. However, few are known extent to which individuals adhere to recommended social distancing practice and factors associated with it. Therefore, this study aimed to assess social distancing practice and associated factors in response to COVID-19 in Bule Hora town, southern Ethiopia 2020.

## Materials and methods

### Study setting, *design and period*

A community based cross-sectional study was conducted in Bule Hora town, West Guji Zone, Oromia Region from September 15–30, 2020. Bule Hora town is the capital of West Guji Zone, located 467 km South to Addis Ababa, Ethiopia's capital. Administratively the town has 4 kebeles (i.e. Smallest administrative unit in Ethiopia). The estimated number of the households in the town is 11, 766.

**Study population, sample size determination and procedure.**   All households of Bule Hora town were source population. Whereas, all randomly selected households within the town during the data collection period were study population. All adult household members aged 18 years and above were included in the study. Critically ill and adults who lived less than six months in the town were excluded from the study. The sample size for the first objective was determined by using single population proportion formula. Considering the proportion of social distancing of 35.3% obtained from previous study conducted in Bangladesh [23], 95% confidence interval (Z = 1.96) and 5% margin of error (d). Then, by substituting the aforementioned figure in to the single population proportion sample size calculation formula, the calculated sample size became 351. Sample size for second objective (identification of factors associated with social distancing practice) was computed by Epi info7 Statcalc version 7.1.4.0

software with the assumptions of, 95% level of confidence, power of 80%, the ratio of exposed to unexposed 1:1 and percent of outcome in unexposed group 64.2% and AOR of 1.9. The percent of outcome in unexposed group and AOR were taken from the study conducted in UK [24]; the determinate variable was (Age 18–34 years). Then required sample for the second objective became 406. Therefore among the two sample sizes calculated, the largest sample size was obtained from the second objective. Then after adding 10% non-response rate to 406, the final minimum total sample size became 447. First all the four kebeles (lowest administrative unit) of the town administration were included in the study. Then after getting the number of households from the town administration Office, the calculated sample size were allocated proportional to the size of population in each kebele. Subsequently, Simple random sampling technique was used to select the households from each kebele. Within selected households, adults (at least 18 years) old were interviewed. In case of presence of more than one eligible adult in the household, lottery method was used to select one adult for the interview.

**Data collection procedures and quality assurance.** The data were collected by a pretested structured interviewer-administered questionnaire. The questions which assess the level of compliance to social distancing practice and associated factors were adapted from previous peer reviewed articles, WHO and FMOH guidelines [10,11,23–25]. The adapted questions were modified and contextualized to fit the local situation and the research objectives. Primarily the questionnaire was prepared in English (**S1 File**) and then translated to the local language "Afaan Oromo" by fluent speakers of both language and then translated back to English to keep the consistency of the questionnaire. The questionnaire contains socio-demographic characteristics, chronic medical conditions, risk perceptions towards COVID-19, knowledge and attitude towards social distancing practices for the prevention of COVID 19, Social distancing practice related questions.

The knowledge level about social distancing practice and COVID-19 was assessed using "Yes' or "No" questions. Five point Likert scale was used to assess attitude (5 = Strongly Agree, 4 = agree, 3 = neutral, 2 = disagree and 1 = strongly disagree) related to social distancing and COVID-19. Whereas, a three point Likert scale was used to assess social distancing practice of the participants (2 = Always, 1 = Occasional, 0 = Never). Two days training was given for data collectors and supervisors on data collection tools and procedures. During data collection personal protective equipment like sanitizer, face mask and glove were secured for each data collectors and supervisors. Questionnaire was pretested on 5% of expected sample size (n = 22) at Gerba town, one week prior to data collection to check whether the questionnaire was accurate. No adjustment was necessary. The overall supervision was carried out by investigators during data collection period on daily basis and data were cleared and checked daily its completeness and consistency before processing and analysis. During data collection a participant having clinical features related to COVID-19 were screened by digital thermometer. But no one has been identified with high grade fever. All the study participants were encouraged to participate in the study voluntarily and at the same time they were also told that they have the right not to participate.

## Study variables and operational definition

The dependent variable of the study was social distancing practice and the explanatory variables were **Socio-demographic factors (**sex, age, residence, income, religion, educational status, marital status, occupation, household tenure and family size), **knowledge** and **attitude** towards social distancing for the responses of COVID-19, **Risk perceptions** towards COVID-19 and **Chronic medical history.**

- **Knowledge:** Participants who answered ≥50% of correct answers among the total knowledge related questions were classified as having a good knowledge. Whereas participants who answered < 50% of the questions were classified as having poor knowledge.

- **Attitude:** Participants who answered ≥50% of correct answers among the total attitude related questions were classified as having a positive attitude. Whereas, participants who answered <50% of questions were classified as having negative attitude.

- **Social distancing practice:** Eight questions with a three Likert scale were collected and the total social distancing practice score was calculated by summing the Likert score. Thus, **p**articipants who answered ≥50% of correct answers among the total **eight** social distancing practice related questions were regarded as having ***good practice***. Whereas, participants who answered less than 50% of the questions were taken as having a ***poor practice.***

- **Perceived susceptibility:** is how likely one considered oneself (his/her families) would be infected with COVID-19 if no preventive measure was taken. Hence, Participants who scored ≥50% of questions were categorized as having high perceived susceptibility. Whereas, participants who scored <50% of questions were categorized as having low perceived susceptibility of contracting COVID-19.

- **Perceived seriousness:** is perceived chance of having COVID-19 cure and survival if infected with COVID-19. Participants who scored ≥50% of questions were categorized as having high perceived severity. Whereas, participants who scored <50% of questions were categorized as having low perceived severity of COVID-19.

- **Perceived self-efficacy:** A person's belief in his or her ability to practice social distancing practice. Participants who scored ≥50% of questions were categorized as having high perceived self-efficacy. Whereas, participants who scored <50% of questions were categorized as having low perceived self-efficacy of practicing social distancing.

- **Perceived Benefits:** is perceived benefits of practicing social distancing for the prevention of COVID-19. Participants who scored ≥50% of questions were categorized as having high perceived benefits from practicing social distancing. Whereas, participants who scored <50% of questions were categorized as having low perceived benefits of practicing social distancing.

- **Perceived Barriers**: Perceived barriers to social distancing practice as a preventive measure of COVID-19. Participants who scored ≥50% of questions were categorized as having high perceived barriers. Whereas, participants who scored <50% of questions were categorized as having low perceived barriers to measures of social distancing practice.

**Data processing and analysis.** The collected data were cleaned, coded, and entered by Epi-DATA version 3.5 and exported to statistical package for social science (SPSS) version 23.0 for analysis. Median with Inter quartile range (IQR) was used to summarize quantitative variable. The results were presented by tables, figures and different interactive charts. Binary logistic regression analysis was done to examine statistical association between social distancing practices and independent variables. Variables with p-value <0.25 on bivariate analysis were further entered into multivariable logistic regression to identify statistically significant variables. The multicollinearity between independent variables was checked by using variation inflation factor (VIF) and tolerance test. The Hosmer-Lemeshow test was done to check the model fitness for analysis. A reliability analysis of the questionnaires was checked and Cronbach's alpha showed the questionnaire were passed the acceptable reliability number (α = 0.82). Adjusted odds ratios (AOR) together with 95% CI were used to estimate the strength of associations and statistical significance was declared at a p-value < 0.05 (S2 File).

**Ethical considerations.** Primarily the study protocol was officially approved by the Research and Publication Directorate of Bule Hora University (Ref.No: BHU/RPD/270/13). Based on the approval, an official letter was written by RPD to Bule Hora Town

Health office and Bule Hora Town Administration office. The Town health office wrote the letter to respective kebeles for cooperation. At last the data were collected after assuring the confidentiality nature of responses and obtaining verbal consent from the study participant.

## Results

### Socio-demographic characteristics of study participants

Out of the total of 447 sampled participants 410 of them were voluntarily interviewed and make the response rate of 91.7%. The median (±IQR) age of study participants was 28(±9) years of age. The majority, 129(31.5%) of study participants were found in the age group of 26–30 years. More than half, 223 (54.4%) of the participants were female. Likewise, nearly, three-fourth of the participants was married. About 142 (34.6%) of the participants have no formal education. Nearly one-fifth, 92 (22.4) of the study participants were government employed. Two hundred seven, (50.5%) and 132(32.3%) of participants have TV and Radio respectively. Concerning housing tenure, 252 (61.5%) of respondents were living in rental houses. Pertaining the family size more than two-fifth, 176 (43%) of the participants have a family size of ≥5. The majority of the study participants, 146 (35.6%) have a monthly income of ≤1000 (Table 1).

### Chronic medical condition and behavior of the study participants

Regarding the participant's chronic medical condition and behavioral history about quarter, 99(24.1%) of the study participants have at least one type of chronic medical history. Twenty five (6.1%), 32 (7.8%) and 24 (5.9%) of participants have DM, hypertension and asthmatic problems respectively. On the other hand small proportion, 28(6.8%) of the study participants are smoke cigarette (Table 2).

### Risk perceptions of the study participants about social distancing practice perceived susceptibility

Of the total more than half, 232 (56.6%) of the respondents were strongly disagreed that there is less chance to transmit infection to family members from sick person (Table 3).

**Perceived severity.** Out of the total respondents, 148(36.1%) agreed that COVID-19 will be more serious among elderly and people with comorbidities. Majority of the respondents agreed that if they were infected with COVID-19, they will suffer severe symptoms (Table 4).

**Perceived self-efficacy.** Out of the total respondents, 123(30%) were much confident that they can get access to the reliable health information on COVID-19. About one hundred nineteen (29%) of respondents were much confident that they will eat healthy diet to prevent covid-19(Table 5).

**Perceived barriers.** Concerning the perceived barriers of respondents, about one-third 136(33.2) of them were strongly agreed that it is hard refraining social gatherings at one's home. One hundred fifty one (36.8%) of respondents were strongly agreed that it is hard to stay home too much (Table 6).

**Perceived benefits.** Concerning the perceived benefits of respondents, about 191(46.6%) of them were agreed that doing protective measures of covid-19 is caring for themselves and their families. Nearly one-third, 142(34.6%) of respondents were greed that keeping social distancing is setting good example for others19 (Table 7).

**Table 1. Socio-demographic characteristics of study participants at Bule Hora town, 2020.**

| Variable | Category | Frequency | Percent % |
|---|---|---|---|
| Age | ≤20 | 46 | 11.2 |
| | 21–25 | 92 | 22.4 |
| | 26–30 | 129 | 31.5 |
| | 31–35 | 78 | 19 |
| | >35 | 65 | 15.9 |
| Sex of respondent | Male | 187 | 45.6 |
| | Female | 223 | 54.4 |
| Marital status | Married | 304 | 74.1 |
| | Single | 84 | 20.5 |
| | Divorced | 11 | 2.7 |
| | Widowed | 11 | 2.7 |
| Educational status | No formal education | 142 | 34.6 |
| | Primary completed | 50 | 12.2 |
| | Secondary completed | 112 | 27.3 |
| | Higher and above | 106 | 25.9 |
| Occupational status | Government employed | 92 | 22.4 |
| | Merchant/Trade | 76 | 18.5 |
| | Farmer | 37 | 9 |
| | Private | 79 | 19.3 |
| | Housewife | 90 | 22 |
| | Others | 36 | 8.8 |
| Housing tenure | Private | 158 | 38.5 |
| | Rental | 252 | 61.5 |
| Television | Yes | 207 | 50.5 |
| | No | 203 | 49.5 |
| Radio | Yes | 132 | 32.3 |
| | No | 277 | 67.7 |
| Family size | ≤2 | 78 | 19 |
| | 3–4 | 156 | 38 |
| | ≥5 | 176 | 43 |
| Monthly income of respondents in Ethiopian birrs* | ≤1000 | 146 | 35.6 |
| | 1001–3000 | 132 | 32.2 |
| | 3001–5000 | 63 | 15.4 |
| | ≥5001 | 69 | 16.8 |

*1 Birr = 0.0229$.

### Study participants' knowledge about risky groups, symptoms, prevention methods of COVID-19

Overall, 222(54.1%) [95% CI (49.3, 59.2%)] of the study participants have good knowledge towards COVID-19 and its prevention methods. Four hundred eight, (99.5%) of respondents ever heard about corona virus. Majority, 157 (38.5%) of respondents obtained information regarding COVID-19 from health personnel. One hundred sixty six (40.5%) of participants claimed that health personnel is trusted source of information. More than half, 246 (60%) of respondents mentioned that the main causes of COVID-19 is virus (Table 8).

**Table 2. Medical condition of the study participants at Bule Hora town, 2020.**

| Variable | Category | Frequency | Percent % |
|---|---|---|---|
| Do you have DM | Yes | 25 | 6.1 |
| | No | 288 | 70.2 |
| | I don't know | 97 | 23.7 |
| Do you have HTN | Yes | 32 | 7.8 |
| | No | 286 | 69.8 |
| | I don't know | 92 | 22.4 |
| Do you have Cardiac problem | Yes | 9 | 2.2 |
| | No | 313 | 76.3 |
| | I don't know | 88 | 21.5 |
| Do you have Asthma | Yes | 24 | 5.9 |
| | No | 318 | 77.6 |
| | I don't know | 68 | 16.6 |
| Do you have Cancer | Yes | 1 | 0.2 |
| | No | 318 | 77.6 |
| | I don't know | 91 | 22.2 |
| Do you have HIV/AIDS | Yes | 28 | 6.8 |
| | No | 247 | 60.2 |
| | I don't know | 135 | 32.9 |
| Do you smoke cigarette | Yes | 28 | 6.8 |
| | No | 382 | 93.2 |

## Attitudes of study participants about COVID-19 and social distancing

Out of the total respondents, 298(72.7% [95% CI (68.8, 76.6%)] of the study participants have positive attitude towards the social distancing practices for the prevention of COVID-19. Of the total study participants, 234(57.1%) were strongly disagreed to stay at home for certain period (14 days) to prevent covid-19 spread if government will order so. Nearly one-third, 125 (30.5%) of the respondents were agreed that social distancing can prevent covid-19 spread (Table 9).

## Social distancing practice of study participant for COVID-19 prevention

In this study nearly two-in-five, 157 (38.3%) [95% CI (33.5, 43.1%)] of the study participants have good social distancing practices for the prevention of COVID-19. Out of total respondents, 169(41.2%) always avoided contact with someone who is displaying symptoms of coronavirus. Two hundred fifty six, (62.4%) of respondents never avoided non-essential use of

**Table 3. Perceived susceptibility of study participants toward COVID-19 pandemic.**

| Perceived susceptibility | Responses | | | | |
|---|---|---|---|---|---|
| | S. disagree No (%) | Disagree No (%) | Neutral No (%) | Agree No (%) | S. agree No (%) |
| Less chance to transmit infection to family members from sick person? | 232 (56.6) | 96 (23.4) | 26(6.3) | 42(10.2) | 14(3.4) |
| No chance to get infection for healthy person | 157(38.3) | 157(38.3) | 34(8.3) | 46(11.2) | 16(3.9) |
| Little chance to get infection for young | 128(31.2) | 178(43.4) | 46(11.2) | 40(9.8) | 18(4.4) |
| High chance to get infection from foreigner | 81(19.8) | 122(29.8) | 70(17.1) | 108(26.3) | 29(7.1) |
| Easily get disease in crowded place | 50(12.2) | 94(22.9) | 61(14.9) | 165(40.2) | 40(9.8) |
| Healthy life style will reduce the chance of infection | 39(9.5) | 101(24.6) | 61(14.9) | 163(39.8) | 46(11.2) |

**Table 4. Perceived severity of study participants toward COVID-19 pandemic.**

| Perceived severity | Responses | | | | |
|---|---|---|---|---|---|
| | S. disagree No (%) | Disagree No (%) | Neutral No (%) | Agree No (%) | S. agree No (%) |
| COVID-19 will be more serious among elderly and people with comorbidities? | 56(13.7) | 54(13.2) | 50(12.2) | 148(36.1) | 102(24.9) |
| If I were infected with COVID-19, I will suffer severe symptoms | 51(12.4) | 42(10.2) | 67(16.3) | 180(43.9) | 70(17.1) |
| If I were infected with COVID-19, I could not survive | 50(12.2) | 68(16.6) | 101(24.6) | 120(29.3) | 70(17.1) |
| I can suffer from COVID-19 without signs and symptoms | 49(12) | 74(18) | 100(24.4) | 137(33.4) | 48(11.7) |
| COVID-19 will be treated if I were infected | 35(8.5) | 67(16.3) | 102(24.9) | 151(36.8) | 55(13.4) |
| If I were infected with COVID-19, i will recover spontaneously | 40(9.8) | 78(19) | 99(24.1) | 125(30.5) | 68(16.6) |

public transport when possible. Majority, 278(67.8%) of respondents never work at home (Table 10).

## Factors associated with knowledge level of respondents towards COVID-19

The out puts of the bi-variable and multivariable logistic regression analyses of factors associated with knowledge level of the participant's found that, being employed were 65% more likely to have good knowledge regarding the prevention measures of COVID-19 compared to unemployed respondents [**AOR** = 1.65(1.05–2.58)]. Similarly, respondents who had positive attitude were 65% more likely of having a good knowledgeable than respondents who had negative attitude [AOR = 1.65(1.02–2.66)]. Respondents who had low perceived susceptibility were 35% less likely to have good knowledge than their counter part [AOR = 0.65(0.43–0.99)] (Table 11).

## Factors associated with attitudes of study participants about COVID-19 and social distancing

Bi-variable and multivariable binary logistic regression was used to identify factors associated with the attitude of study participants regarding COVID-19 and social distancing practices. Accordingly, variables which had a p-value of ≤0.25 during bivariable logistic regression were further entered to multivariable binary logistic regression. After adjusting for confounding variables, the odds of positive attitude was 68% [AOR = 0.32(0.13–0.80)] and 66% [AOR = 0.34(0.14–0.82)] reduced among respondents who were in age group of 26–30 and 31–35 years as compared to respondents who were above 35 years of age respectively. Respondents who had perceived less severity and perceived low self-efficacy were 43% [AOR = 0.57 (0.32–0.99)] and 48% [AOR = 0.52(0.31–0.88)] less likely to have positive attitude than their counter parts respectively (Table 12).

**Table 5. Perceived self-efficacy of study participants toward COVID-19 pandemic.**

| Perceived self-efficacy | Responses | | | | |
|---|---|---|---|---|---|
| | No No (%) | Low confident No (%) | Neutral No (%) | Much No (%) | High confident No (%) |
| I can get access to the reliable health information on COVID-19 | 84(20.5) | 95(23.2) | 30(7.3) | 123(30) | 78(19) |
| I will eat healthy diet to prevent COVID-19 | 88(21.5) | 58(14.1) | 43(10.5) | 119(29) | 102(24.9) |
| To prevent COVID-19, I will wash my hands | 61(14.9) | 58(14.1) | 26(6.3) | 130(31.7) | 135(32.9) |
| I can prevent COVID-19 | 39(9.5) | 66(16.1) | 75(18.3) | 148(36.1) | 82(20) |
| To prevent COVID-19, I will avoid visiting crowded places | 42(10.2) | 71(17.3) | 46(11.2) | 160(39) | 91(22.2) |
| To prevent COVID-19, I will use face mask whenever I go to crowded place | 40(9.8) | 62(15.1) | 51(12.4) | 154(37.6) | 103(25.1) |

**Table 6. Perceived barriers of study participants toward COVID-19 pandemic.**

| Perceived barriers | Responses | | | | |
|---|---|---|---|---|---|
| | S. disagree No (%) | Disagree No (%) | Neutral No (%) | Agree No (%) | S. agree No (%) |
| Hard to refrain social gatherings at home | 84(20.5) | 42(10.5) | 26(6.3) | 122(29.8) | 136(33.2) |
| Hard to stay home too much | 49(12) | 50(12.2) | 24(5.9) | 136(33.2) | 151(36.8) |
| Difficult using face mask daily? | 41(10) | 57(13.9) | 25(6.1) | 140(34.1) | 147(35.9) |
| Can't afford to buy soap/alcohol containing hand sanitizer | 27(6.6) | 46(11.2) | 60(14.6) | 156(38) | 120(29.3) |

## Factors associated with social distancing practice for the prevention of Covid-19

During the bivariable binary logistic regression, age of respondents, educational status, having Television, Cigarette smoking, Attitude level, knowledge status, Perceived Susceptibility, Perceived Barriers and Perceived Benefits were statistically significant at a p-value of <0.25 and identified as the candidates for the multivariable binary logistic regression analysis so as to control the potential presence of confounding variables. As a result, at the multivariate model the Age of respondents, occupational status, knowledge status, attitude level and perceived susceptibility were significantly associated with good social distancing practice at p<0.05.

Accordingly, the odds of good social distancing practice was 45% reduced among the household who have negative attitude towards social distancing practices for the prevention of COVID-19 as compared to their counter parts [AOR = 0.55 (95% CI:0.31–0.95)]. Similarly, the odds of good social distancing practices was 41% reduced among the household who have poor knowledge about social distancing practices as compared to their counter parts [AOR = 0.59 (95% CI:0.36–0.95)]. On the other hand, the odds of good social distancing practices was 67% reduced among individuals who have low susceptibility perception for contracting COVID-19 as compared to individuals who have high susceptibility perception of contracting COVID-19 [AOR = 0.33(95%CI: 0.20–0.54)]. Those respondents who were employed were 6 times more likely to comply with social distancing practice as compared to those who were unemployed [AOR = 6.10(95%CI: 3.46–10.74)]. The age of respondents was also positively associated with social distancing practices. The odd of good social distancing practices was 2.5 [AOR = 2.56(95% CI: 1.18–5.54)] and 3.5 [AOR = 3.57(95%CI: 1.56–8.18)] times higher among individuals who are in the age group of 26–30 and 31–35 years as compared to individuals who are above 35 years of age respectively (Table 13).

## Discussion

In this study an overall, 222(54.1%) [95% CI (49.3, 59.2%)] of the study participants have good knowledge towards COVID-19 and its prevention methods. This is similar to a study

**Table 7. Perceived benefits of study participants toward COVID-19 pandemic.**

| Perceived benefits | Responses | | | | |
|---|---|---|---|---|---|
| | S. disagree No (%) | Disagree No (%) | Neutral No (%) | Agree No (%) | S. agree No (%) |
| When I am doing something protective measures of COVID-19, I am caring for myself and my families | 73(17.8) | 31(7.6) | 43(10.5) | 191(46.6) | 72(17.6) |
| When I keep social distancing, I am setting a good example for others | 37(9) | 75(18.3) | 73(17.8) | 142(34.6) | 83(20.2) |
| When I wear face mask at crowded area, I am decreasing my chances of contracting COVID-19? | 28(6.8) | 73(17.8) | 72(17.6) | 159(38.8) | 78(19) |
| Staying home will reduce my chances of contracting COVID-19? | 24(5.9) | 54(13.2) | 76(18.5) | 155(37.8) | 101(24.6) |

**Table 8. Participants' knowledge of risky groups, symptoms, prevention methods of COVID-19 among Bule Hora town adults (n = 410).**

| Variable | Category | Frequency | Percent |
|---|---|---|---|
| Ever heard about corona virus | Yes | 98 | 99.5 |
|  | No | 2 | 0.5 |
| What was your source of information? | Health personnel | 157 | 38.3 |
|  | Social media | 76 | 18.5 |
|  | FMOH sources | 23 | 5.6 |
|  | Mass media | 125 | 30.5 |
|  | Friends/family members/relatives | 29 | 7.1 |
| Trusted source of information | Health personnel | 166 | 40.5 |
|  | Social media | 58 | 14.1 |
|  | FMOH sources | 38 | 9.3 |
|  | Mass media | 128 | 31.2 |
|  | Friends/family members/relatives | 20 | 4.9 |
| The cause of COVID-19 is? | Virus | 246 | 60 |
|  | Others* | 164 | 40 |
| Can COVID-19 transmit human-to-human? | Yes | 386 | 94.1 |
|  | No | 24 | 5.9 |
| What are the modes of transmission of COVID-19? | Airborne | 152 | 37.1 |
|  | Physical contact with contaminated object | 208 | 50.7 |
|  | Physical contact with infected people | 154 | 37.6 |
|  | Eating raw meat | 148 | 36.1 |
| Prevention methods of COVID-19 | Avoid close contact with people who are sick | 237 | 57.8 |
|  | Frequent hand washing with soap and water/alcohol-based hand sanitizer | 322 | 78.5 |
|  | Avoid touching your eye, nose, mouth with unwashed hands | 169 | 41.2 |
|  | Avoid shaking hands | 152 | 37.1 |
|  | Avoid crowded place | 255 | 62.6 |
|  | Disinfecting/cleaning objects and surfaces | 134 | 32.7 |
|  | Stay at home/work at home | 84 | 20.5 |
|  | Practicing good respiratory hygiene | 113 | 27.6 |
| The main clinical symptoms of COVID-19 | Fever | 344 | 83.9 |
|  | Dry Cough | 318 | 77.6 |
|  | Breathing difficulty | 182 | 44.4 |
|  | Fatigue | 110 | 26.8 |
|  | Sneezing | 199 | 48.5 |
|  | Headache | 163 | 39.8 |
| There is no effective vaccine for COVID-19? | Yes | 233 | 56.8 |
|  | No | 177 | 43.2 |
| There is no any definitive treatment of COVID-19 currently? | Yes | 278 | 67.8 |
|  | No | 132 | 32.2 |
| High-risk population of COVID-19 | Children | 128 | 31.2 |
|  | Elderly | 179 | 43.7 |
|  | Pregnant women | 72 | 17.6 |
|  | People with chronic disease | 121 | 29.5 |
|  | Cigarette smokers | 43 | 10.5 |

conducted at Northwest Syria (51%) [26]. However, it is lower than studies conducted across the globe: Southern Ethiopia 90% [21], Iran 90% [27], USA 71.7% [28], Nepal 79% [29], Uganda 84% [22], Italy 83.4% [30], Bangladesh 70% [23] and Paraguay's 62% [31]

**Table 9. Attitudes of study participants about COVID-19 and social distancing.**

| Questions | Responses | | | | |
|---|---|---|---|---|---|
| | S. disagree No (%) | Disagree No (%) | Neutral No (%) | Agree No (%) | S. agree No (%) |
| Do you like to stay at home for certain period (14 days) to prevent COVID-19 spread if government will order so? | 234(57.1) | 68(16.6) | 25(6.1) | 66(16.1) | 17(4.1) |
| Do you think that social distancing (e.g. stay 2 m apart, avoiding crowds, etc.) can prevent COVID-19 spread? | 69(16.8) | 101(24.6) | 82(20) | 125(30.5) | 33(8) |
| Do you agree that we should cancel business/recreational trips at this time? | 69(16.8) | 149(36.3) | 101(24.6) | 60(14.6) | 31(7.6) |
| Do you believe that working from home can help to control COVID-19? | 68(16.6) | 109(26.6) | 120(29.3) | 67(16.3) | 46(11.2) |
| When someone has signs and symptoms of COVID-19, I can confidently keep my physical distance from him/her? | 33(8) | 43(10.5) | 75(18.3) | 184(44.9) | 75(18.3) |
| Do you think that, Ethiopia is in a good position to **contain** COVID-19? | 28(6.8) | 80(19.5) | 85(20.7) | 130(31.7) | 87(21.2) |

demonstrated poor knowledge in related to the covid-19. The observed difference might be due to the socio-demographic and period of the study commencement in which most of the studies were conducted immediate to the pandemic. On the other hand it is higher than the magnitude reported from Thailand where, the majority, 73.4% of the study participants had poor knowledge of COVID-19 prevention and control [32]. Majority, 157 (38.5%) of respondents obtained information regarding COVID-19 from health personnel. One hundred sixty six (40.5%) of participants claimed that health personnel is trusted source of information. This is consistent with a study conducted at Kenya [33]. More than half, 246 (60%) of respondents mentioned that the main causes of COVID-19 is virus. More than three-in-five (62.6%) and more than three-in-fourth (78.5%) of the participants were responded that avoiding crowded place and frequent hand washing with soap and water/alcohol-based hand sanitizer as the main prevention methods of COVID-19.

Our study also identified factors affecting the knowledge level of participants about the coronavirus infection and its prevention mechanism. In our study the knowledge level of the participant was significantly higher among employed participants, who had positive attitude and those who had high perceived susceptibility of contracting the corona infection. This result is consistent with studies conducted in Southern Ethiopia [21] and Egypt [34] and China [25] in which participants with high socioeconomic status and hold optimistic attitudes were more knowledgeable about COVID-19.This might be associated with being employed has improved the prospect of sharing and seeking updated information about the COVID-19

**Table 10. Social distancing practice of study participant for COVID-19 prevention.**

| Questions | Responses | | |
|---|---|---|---|
| | Always No (%) | Occasional No (%) | Never No (%) |
| Avoid contact with someone who is displaying symptoms of coronavirus | 169(41.2) | 154(37.6) | 87(21.2) |
| Avoid non-essential use of public transport when possible | 132(32.2) | 22(5.4) | 256(62.4) |
| Work at home | 121(29.5) | 11(2.7) | 278(67.8) |
| Avoid large and small gatherings in public spaces (pubs, restaurants, leisure centers) | 163(39.8) | 7(1.7) | 240(58.5) |
| Avoid gatherings with friends and family | 135(32.9) | 19(4.6) | 256(62.4) |
| Maintaining non-contact greetings | 348(84.9) | 22(5.4) | 40(9.8) |
| Maintain 2 meters distance between yourself & other people | 156(38) | 40(9.8) | 214(52.2) |
| Stay home when ill | 91(22.2) | 88(21.5) | 231(56.3) |

**Table 11. Factors associated with knowledge of risky groups, symptoms, and prevention methods of COVID-19 among households of Bule Hora town, Southern Ethiopia, 2020.**

| Variables | | Knowledge | | | | COR (95% Cl) | AOR (95% Cl) |
|---|---|---|---|---|---|---|---|
| | | Good | | Poor | | | |
| | | No | % | No | % | | |
| Sex of respondent | | | | | | | |
| | Male | 109 | 58.3 | 78 | 41.7 | 1.36 (0.91–2.01) | 1.25 (0.82–1.90) |
| | Female | 113 | 50.7 | 110 | 49.3 | 1 | 1 |
| Age of respondents | | | | | | | |
| | ≤20 | 21 | 45.7 | 25 | 54.3 | 0.63(0.27–1.35) | |
| | 21–25 | 48 | 52.2 | 44 | 47.8 | 0.82(0.43–1.56) | |
| | 26–30 | 69 | 53.5 | 60 | 46.5 | 0.87(0.47–1.58) | |
| | 31–35 | 47 | 60.3 | 31 | 39.7 | 1.14(0.58–2.23) | |
| | >35 | 37 | 56.9 | 28 | 43.1 | 1 | |
| Educational status | | | | | | | |
| | No formal education | 76 | 53.5 | 66 | 46.5 | 1 | |
| | Primary completed | 25 | 50 | 25 | 50 | 0.86(0.45–1.65) | |
| | Secondary completed | 61 | 54.5 | 51 | 45.5 | 1.03(0.63–1.70) | |
| | College and above | 60 | 56.6 | 46 | 43.4 | 1.13(0.68–1.87) | |
| Occupational status | | | | | | | |
| | Employed# | 81 | 62.8 | 48 | 37.2 | 1.67(1.09–2.56) | 1.65(1.05–2.58)** |
| | Unemployed | 141 | 49.8 | 140 | 50.2 | 1 | 1 |
| TV | | | | | | | |
| | Yes | 118 | 57 | 89 | 43 | 1 | 1 |
| | No | 104 | 51.2 | 99 | 48.8 | 0.79(0.53–1.16) | 0.86(0.57–1.30) |
| Radio | | | | | | | |
| | Yes | 77 | 58.3 | 55 | 41.7 | 1 | |
| | No | 144 | 52 | 133 | 48 | 0.77(0.50–1.17) | |
| Family size | | | | | | | |
| | ≤2 | 38 | 47.7 | 40 | 42.3 | 0.67(0.39–1.15) | 0.71(0.41–1.26) |
| | 3–4 | 81 | 51.9 | 75 | 48.1 | 0.76(0.49–1.18) | 0.77(0.48–1.21) |
| | ≥5 | 103 | 58.5 | 73 | 41.5 | 1 | 1 |
| At least one Chronic disease | | | | | | | |
| | Yes | 60 | 60.6 | 39 | 39.4 | 1 | 1 |
| | No | 162 | 52.1 | 49 | 47.9 | 0.70(0.44–1.12) | 0.80(0.48–1.34) |
| Attitude | | | | | | | |
| | Negative | 51 | 45.5 | 61 | 54.5 | 1 | 1 |
| | Positive | 171 | 57.4 | 127 | 42.6 | 1.16(1.04–2.49) | 1.65(1.02–2.66)* |
| Perceived Susceptibility | | | | | | | |
| | Low susceptibility | 83 | 48.5 | 88 | 51.5 | 0.67(0.45–1.07) | 0.65(0.43–0.99)* |
| | Highly susceptibility | 139 | 58.2 | 100 | 41.8 | 1 | 1 |
| Perceived Severity | | | | | | | |
| | Less severe | 45 | 57 | 34 | 43 | 1.17(0.71–1.92) | |
| | Highly Severe | 174 | 53 | 154 | 47 | 1 | |
| Perceived self-efficacy | | | | | | | |
| | Low self-efficacy | 63 | 57.8 | 46 | 42.2 | 1.23(0.78–1.90) | |
| | High self-efficacy | 159 | 52.8 | 142 | 47.2 | 1 | |
| Perceived Benefits | | | | | | | |
| | Not Benefits | 26 | 45.6 | 31 | 54.4 | 0.67(0.38–1.17) | |

(*Continued*)

**Table 11.** (Continued)

| Variables | | Knowledge | | | | COR (95% Cl) | AOR (95% Cl) |
|---|---|---|---|---|---|---|---|
| | | Good | | Poor | | | |
| | | No | % | No | % | | |
| | Benefits | 196 | 55.5 | 157 | 44.5 | 1 | |

*p-value <0.05,

** p-value <0.001,

***p-value<0.0001;

# Government and private employed.

and its prevention mechanisms. Moreover, a study conducted at Egypt also revealed result which supported our study [34].

The prevalence of Positive attitude, 298(72.7%) [95%CI:68.8%-76.6%] found in this study is comparable to a study done at Bangladeshi [23] and Paraguay's [31] reported a desired attitude of the population towards COVID-19. However, the result is lower than studies conducted in Ethiopia [20,21]. This might be the difference in study participant's characteristics, where the above studies majorities of the respondents were governmental employers. Pertaining to the factors affecting the attitude status of respondents, the odds of positive attitude was reduced among respondents who were in age group of 26–30 and 31–35 years as compared to respondents who were above 35 years of age. Likewise, respondents who had less perceived severity and low perceived self-efficacy were less likely to have positive attitude than their counter parts. This is congruent to a study report done at Addis Ababa, Ethiopia [20] and study conducted at Brazil on Health belief model for coronavirus infection risk determinants [35]. The possible explanation might be due to the reason that having perception of greater severity may lead the community to seek health services earlier.

In this study, 38.3% **[95% CI: 33.5%, 43.1%)]** of the study participants have good social distancing practices for the prevention of COVID-19. This result is supported by a study done in Thailand [32], Bangladesh [23], Kenya [33] and United Kingdom [24]. However, the result is incomparable or lower than studies conducted at Uganda [22]. The observed difference might be due to the difference in timing of the study, the distribution of the outbreak across the nation cities or towns and socio-demographic characteristics of the participants. Moreover, this study was conducted at the time where different governmental sanctions were lifted off; particularly state of emergency was completely removed.

Concerning the social distancing; less than half, 47.8% of respondents maintained 2 meters distance between themselves & other people. This is higher than studies conducted in Northwest Syria, 17% [26] and Nigeria, 20.4% [36]. The possible explanation could due the difference in socio-demographic characteristics and study period. However, our study result is lower than a study conducted at Addis Ababa, Ethiopia where, 89.5% of respondents practiced physical distancing [20], Italy 85.6% [37], Southern Ethiopia 65% [21]. The possible explanation for this lower practice could be due to the participant's behavior of adopting the newly introduced rules and regulations. Furthermore, the participant's natures in the aforementioned studies were urban compared to our study participants. So, this might contributes for the observed lower social distancing practices in our study.

On the other hand, the study result revealed that nearly three in-five (62.4%) of respondents never avoided non-essential use of public transport. This is comparable to study done in Iran (61.8%) [27], South Korea [38] and Addis Ababa, Ethiopia [20]. In our study more than five in-six, (84.9%) of respondents maintained non-contact greeting. This is comparable to a Addis

**Table 12. Factors associated with attitudes of study participants about COVID-19 and social distancing among Households of Bule Hora town, Southern Ethiopia, 2020.**

| Variables | | Attitude | | | | COR (95% Cl) | AOR (95% Cl) |
|---|---|---|---|---|---|---|---|
| | | Positive | | Negative | | | |
| | | No | % | No | % | | |
| Sex of respondent | | | | | | | |
| | Male | 137 | 73.3 | 50 | 26.7 | 1.05 (0.68–1.63) | |
| | Female | 161 | 72.2 | 62 | 27.8 | 1 | |
| Age of respondents | | | | | | | |
| | ≤20 | 34 | 73.9 | 12 | 26.1 | 0.57(0.22–1.45) | 0.41(0.12–1.33) |
| | 21–25 | 67 | 72.8 | 25 | 27.2 | 0.54(0.24–1.20) | 0.43(0.15–1.20) |
| | 26–30 | 91 | 70.5 | 38 | 29.5 | 0.48(0.23–1.03) | 0.32(0.13–0.80)* |
| | 31–35 | 52 | 66.7 | 26 | 33.3 | 0.40(0.18–0.90) | 0.34(0.14–0.82)* |
| | >35 | 54 | 83.1 | 11 | 16.9 | 1 | 1 |
| Educational status | | | | | | | |
| | No formal education | 101 | 71.1 | 41 | 28.9 | 1 | 1 |
| | Primary completed | 40 | 80 | 10 | 20 | 1.64(0.74–3.55) | 1.73(0.73–4.11) |
| | Secondary completed | 78 | 69.6 | 34 | 30.4 | 0.93(0.54–1.60) | 1.27(0.67–2.41) |
| | College and above | 79 | 74.5 | 27 | 23.5 | 1.18(0.67–2.09) | 1.56(0.83–2.92) |
| Occupational status | | | | | | | |
| | Employed# | 89 | 69 | 40 | 31 | 0.76(0.48–1.21) | |
| | Unemployed | 209 | 25.6 | 72 | 74.4 | 1 | |
| Family size | | | | | | | |
| | ≤2 | 47 | 60.3 | 31 | 39.7 | 0.55(0.31–0.97) | 0.70(0.33–1.50) |
| | 3–4 | 122 | 78.2 | 34 | 21.8 | 1.30 (0.78–2.16) | 1.63(0.87–3.06) |
| | ≥5 | 129 | 73.3 | 47 | 26.7 | 1 | 1 |
| At least one Chronic disease | | | | | | | |
| | Yes | 71 | 71.7 | 28 | 28.3 | 1 | |
| | No | 227 | 73 | 84 | 27 | 1.06(0.64–1.76) | |
| Perceived Susceptibility | | | | | | | |
| | Low susceptibility | 119 | 69.6 | 52 | 30.4 | 0.76(0.49–1.18) | 1.06(0.65–1.74) |
| | Highly susceptibility | 179 | 74.9 | 60 | 23.1 | 1 | 1 |
| Perceived Severity | | | | | | | |
| | Less severe | 46 | 58.2 | 33 | 41.8 | 0.44(0.26–0.73) | 0.57(0.32–0.99)* |
| | Highly Severe | 249 | 75.9 | 79 | 24.1 | 1 | 1 |
| Perceived self-efficacy | | | | | | | |
| | Low self-efficacy | 68 | 62.4 | 41 | 37.6 | 0.51(0.32–0.81) | 0.52(0.31–0.88)* |
| | High self-efficacy | 230 | 76.4 | 71 | 23.6 | 1 | 1 |
| Perceived Barriers | | | | | | | |
| | Barriers | 47 | 68.1 | 22 | 31.9 | 0.76(0.43–1.34) | |
| | Not barriers | 251 | 73.6 | 90 | 26.4 | 1 | |

*p-value <0.05,

** p-value <0.001,

***p-value<0.0001;

# Government and private employed.

Ababa, Ethiopia [20]. The possible explanation for this high practice could be the participant's knowledge on the mode of transmission of the disease.

**Table 13. Factors associated with social distancing practices for the prevention of COVID-19 among households of Bule Hora town, Southern Ethiopia, 2020.**

| Variables | | Practice | | | | COR (95% Cl) | AOR (95% Cl) |
|---|---|---|---|---|---|---|---|
| | | Good | | Poor | | | |
| | | No. | (%) | No. | (%) | | |
| Sex of respondent | | | | | | | |
| | Male | 73 | 46.5 | 114 | 45.1 | 0.94 (0.63–1.40) | |
| | Female | 84 | 53.5 | 139 | 54.9 | 1 | |
| Age of respondents | | | | | | | |
| | ≤20 | 16 | 10.2 | 30 | 11.9 | 1.50(0.66–3.42) | 2.26(0.81–6.34) |
| | 21–25 | 36 | 22.9 | 56 | 22.1 | 1.81(0.90–3.63) | 2.04(0.87–4.77) |
| | 26–30 | 51 | 32.5 | 78 | 30.8 | 1.84(0.95–3.55) | 2.56(1.18–5.54)* |
| | 31–35 | 37 | 23.6 | 41 | 16.2 | 2.54(1.25–5.18) | 3.57(1.56–8.18)** |
| | >35 | 17 | 10.8 | 48 | 19.0 | 1 | 1 |
| Educational status | | | | | | | |
| | No formal education | 45 | 28.7 | 97 | 38.3 | 1 | 1 |
| | Primary completed | 23 | 14.6 | 27 | 10.7 | 1.83(0.95–3.54) | 1.80(0.81–4.00) |
| | Secondary completed | 43 | 27.4 | 69 | 27.3 | 1.34(0.79–2.25) | 1.45(0.76–2.97) |
| | College and above | 46 | 29.3 | 60 | 23.7 | 1.65(0.98–2.78) | 0.65(0.33–1.28) |
| Occupational status | | | | | | | |
| | Employed# | 80 | 51.8 | 49 | 19.4 | 4.32(2.78–6.72) | 6.10(3.46–10.74)*** |
| | Unemployed | 107 | 68.2 | 174 | 68.8 | 1 | 1 |
| Housing tenure | | | | | | | |
| | Private | 58 | 36.9 | 100 | 39.5 | 1 | |
| | Rental | 99 | 63.1 | 153 | 60.5 | 1.11(0.74–1.68) | |
| TV | | | | | | | |
| | Yes | 86 | 54.8 | 121 | 47.8 | 1 | 1 |
| | No | 71 | 45.2 | 132 | 52.2 | 0.75(0.50–1.12) | 0.65(0.40–1.05) |
| Radio | | | | | | | |
| | Yes | 54 | 34.4 | 78 | 31 | 1 | |
| | No | 103 | 65.6 | 174 | 69 | 0.85(0.56–1.30) | |
| Family size | | | | | | | |
| | ≤2 | 27 | 17.2 | 51 | 20.2 | 0.88(0.50–1.54) | |
| | 3–4 | 64 | 40.8 | 92 | 36.4 | 1.15(0.74–1.80) | |
| | ≥5 | 66 | 42 | 110 | 43.5 | 1 | |
| Cigarette smoking | | | | | | | |
| | Yes | 14 | 8.9 | 14 | 5.5 | 1 | 1 |
| | No | 143 | 91.1 | 239 | 94.5 | 0.59(0.27–1.29) | 0.84(0.34–2.06) |
| At least one Chronic disease | | | | | | | |
| | Yes | 36 | 22.9 | 63 | 24.9 | 1 | |
| | No | 121 | 77.1 | 190 | 75.1 | 1.14(0.69–1.78) | |
| Attitude | | | | | | | |
| | Positive | 123 | 78.3 | 175 | 69.2 | 1 | 1 |
| | Negative | 34 | 21.7 | 78 | 30.8 | 0.60(0.39–0.98) | 0.55(0.31–0.95)* |
| Knowledge | | | | | | | |
| | Good | 102 | 65 | 120 | 47.4 | 1 | 1 |
| | Poor | 55 | 35 | 133 | 52.6 | 0.48(0.32–0.73) | 0.59(0.36–0.95)* |
| Perceived Susceptibility | | | | | | | |
| | Low susceptibility | 42 | 26.7 | 129 | 51 | 0.35(0.37–0.84) | 0.33(0.20–0.54)*** |
| | Highly susceptibility | 115 | 73.3 | 124 | 49 | 1 | 1 |

(*Continued*)

**Table 13.** (Continued)

| Variables | | Practice | | | | COR (95% Cl) | AOR (95% Cl) |
|---|---|---|---|---|---|---|---|
| | | Good | | Poor | | | |
| | | No. | (%) | No. | (%) | | |
| Perceived Severity | | | | | | | |
| | Not severe | 34 | 21.9 | 45 | 17.9 | 1.29(0.78–2.12) | |
| | Severe | 121 | 78.1 | 207 | 82.1 | 1 | |
| Perceived Self-efficacy | | | | | | | |
| | Low self-efficacy | 42 | 26.8 | 67 | 26.5 | 1.04(0.64–1.59) | |
| | High self-efficacy | 115 | 73.2 | 186 | 73.5 | 1 | |
| Perceived Barriers | | | | | | | |
| | Barriers | 20 | 12.7 | 49 | 19.4 | 0.68(0.34–1.06) | 0.55(0.28–1.07) |
| | Not barriers | 137 | 87.3 | 204 | 80.6 | 1 | 1 |
| Perceived Benefits | | | | | | | |
| | Not Benefited | 38 | 24.2 | 42 | 16.6 | 0.53(0.28–0.99) | 0.88(0.42–1.87) |
| | Benefits | 119 | 75.8 | 211 | 83.4 | 1 | 1 |

*p-value <0.05,

** p-value <0.001,

***p-value<0.0001;

# Government and private employed.

The odds of good social distancing practice was 45% reduced among household members who have negative attitude towards social distancing practices compared to their counter parts. This finding is supported by other studies conducted in Brazil [39], Hong Kong, China [40] and Bangladesh [23]. This is because an individuals who have positive attitude may have better social distancing practice for COVID-19 preventive measures than the individual who have negative attitude. Respondents who have poor knowledge about social distancing practices have 41% reduced social distancing practices as compared to respondents who have good knowledge about the social distancing practices. This finding is supported by the studies carried out in Hubei, China [25], and Pakistan [41]. This is due to the fact that, respondents who have knowledge on COVID-19 cause, mode of transmission, symptoms and prevention methods would be more likely to practice social distancing.

Odds of good social distancing practices was 67% reduced among individuals who have low perceived susceptibility of contracting COVID-19 as compared to individuals who have high perceived susceptibility of contracting COVID-19. This finding is corroborated by studies conducted in Hong Kong, China [40], South Korea [38] and worldwide survey [42]. Likewise a large survey conducted in 48 countries also reported similar result [42]. This indicates that the perceived level of personal susceptibility has created fear when seeing hard-hitting emotional messaging. As a result individuals became aware and adhere to social distancing practices to reduce perceived threat. Moreover, current evidence showed that respondents with high behavioral responses found to be practicing social distancing.[38].

Those respondents who were employed were 6 times more likely to adhere to social distancing practice as compared to those who were unemployed. The Similar findings were reported by studies conducted in Addis Ababa, Ethiopia [20], Southern Ethiopia [21], Bangladesh [23], Brazil [43] and Uganda [22]. This could be due to the awareness and daily exposure of information, enforcement of social distancing practices within work environment. Likewise, at the time of the incidence of the pandemic the majorities of organization were permitted their staffs

to work at their home and also reduced the number of staff working on daily basis. This would also by itself minimize the use of public transportation and unnecessary gatherings. More importantly, individuals who were more educated or employed would have a greater tendency to engage in protective behaviors during pandemics.

The age of respondents was also positively associated with social distancing practices. The odd of good social distancing practices was higher among respondents who were in the age group of 26–30 and 31–35 years as compared to respondents who are above 35 years of age. This is consistent with a study conducted at Malaysia reported that those older age were more likely to attend daily religious ceremonies [44]. This similarity might be due to the socio-demographic characteristics of the participants, where in both study area the majority of communities are religious. In contrary, a study conducted at United Kingdom revealed that aged 70 and above had good social distancing practice measures [24]. The observed difference might be difference in demographic size and composition among the study areas.

### Limitations and strength of the study

Among the strengths of our study; first this is the first study conducted at the study area. Second, the study included all the kebeles found in the study town, Third; it is a community based study which enables us to generalize our findings for our source population. Despite its strength, the limitations of our study are: the timing of the study conducted where the attention of the COVID-19 had been decreased. The introduction of social desirability biases particularly on social distancing related variables, and lastly, the cross-sectional nature of the study design does not establish the cause and effect relationship.

### Conclusion

This study results showed that the smaller proportion of the study participants had demonstrated good knowledge, and good social distancing practice. Individuals should abide and implement the information released from regional health bureau and FMOH. Moreover, Bule Hora Town Health Office and West Guji Zone Health Department should give emphasis on providing continues awareness creation so as to lift the knowledge of the community, particularly on the mechanisms of covid-19 prevention techniques due stress on social distancing practices. Although, the results of this study can be used as baseline information for the local, regional and national governments and other stakeholders engaged in the prevention and control of COVID-19, further study should be conducted to get more representative data for the policy makers and triangulate with qualitative to explore other different possible determinant factors.

### Supporting information

**S1 File. English version questionnaire.**
(PDF)

**S2 File. Raw SPSS dataset.**
(ZIP)

### Acknowledgments

We would also like to extend our deepest gratitude to Bule Hora University, Research and Publication Directorate for allowing us to conduct this research. We would also like to give our great appreciation to the data collectors: Mr. Alo Edin and Mr. Angefa Ayele and for all the study participants.

## Author Contributions

**Conceptualization:** Anteneh Fikrie, Elias Amaje.

**Data curation:** Anteneh Fikrie, Elias Amaje.

**Formal analysis:** Anteneh Fikrie, Elias Amaje.

**Funding acquisition:** Anteneh Fikrie, Elias Amaje.

**Investigation:** Anteneh Fikrie, Elias Amaje.

**Methodology:** Anteneh Fikrie, Elias Amaje.

**Project administration:** Anteneh Fikrie, Elias Amaje, Wako Golicha.

**Resources:** Anteneh Fikrie, Elias Amaje, Wako Golicha.

**Software:** Anteneh Fikrie, Elias Amaje, Wako Golicha.

**Supervision:** Anteneh Fikrie, Elias Amaje, Wako Golicha.

**Validation:** Anteneh Fikrie, Elias Amaje, Wako Golicha.

**Visualization:** Anteneh Fikrie, Elias Amaje, Wako Golicha.

**Writing – original draft:** Anteneh Fikrie, Elias Amaje, Wako Golicha.

**Writing – review & editing:** Anteneh Fikrie, Elias Amaje, Wako Golicha.

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
