## [Decision Letter · Decision Letter 0]

8 Jun 2021

PONE-D-21-13624

Social Distancing Practice and Associated Factors in Response to COVID-19 Pandemic at West Guji Zone, Southern Ethiopia, 2020. A Community Based Cross-sectional study

PLOS ONE

Dear Dr. Fikrie,

Thank you for submitting your manuscript to PLOS ONE. After careful consideration, we feel that it has merit but does not fully meet PLOS ONE’s publication criteria as it currently stands. Therefore, we invite you to submit a revised version of the manuscript that addresses the points raised during the review process.

We look forward to receiving your revised manuscript.

Kind regards,

Prasenjit Mitra, MD, MRSB, MIScT, FLS, FACSc, FAACC

Academic Editor

PLOS ONE

Journal Requirements: 

2. Please include additional information regarding the survey or questionnaire used in the study and ensure that you have provided sufficient details that others could replicate the analyses. For instance, if you developed a questionnaire as part of this study and it is not under a copyright more restrictive than CC-BY, please include a copy, in both the original language and English, as Supporting Information

3. Please provide additional details regarding participant consent. In the ethics statement in the Methods and online submission information, please ensure you have specified: 1) whether the ethics committee approved the verbal/oral consent procedure, 2) why written consent could not be obtained, and 3) how verbal/oral consent was recorded. If your study included minors, please state whether you obtained consent from parents or guardians in these cases. If the need for consent was waived by the ethics committee, please include this information.

4. Please upload a copy of Supporting Information S1 File and S2 File Which you refer to in your text on page 15.

Additional Editor Comments (if provided):

1- Study setting (methodology section)

Spelling mistake in line 3

2- Explain the sampling technique in methodology section (no sampling technique)

3- What is the aim of calculating sample size for each objective separately?

4- What is meant by kebele?

5- Why you apply pilot testing on 5% of the sample?

6- Why you consider 50% as a cut off point for dividing the people into good or poor in knowledge, attitude, Social distancing practice, ------etc?

As these are means, you can consider those who have above the mean as good and below the mean as poor

7- More explanation of the data collection tool is needed regarding knowledge, attitude, Social distancing practice, ------etc?

8- Rewriting of the references is needed

9- What is meant by Merchant in the occupational status (should be mentioned)?

10- What is currency of the monthly income?

11- What is the value of asking about housing tenure, TV and Radio in sociodemographic characteristics?

12- In this study, 38.3% [95% CI:33.5%, 43.1%)] of the study participants have good social distancing practices for the prevention of COVID-19.

Where does this finding present in your results?

13- The title of the manuscript should be broadened because social distance practice and associated factors are present in table 10, 11 only

Reviewers' comments:

Reviewer's Responses to Questions

**Comments to the Author**

1. Is the manuscript technically sound, and do the data support the conclusions?

Reviewer #1: No

2. Has the statistical analysis been performed appropriately and rigorously? 

Reviewer #1: Yes

3. Have the authors made all data underlying the findings in their manuscript fully available?

Reviewer #1: Yes

4. Is the manuscript presented in an intelligible fashion and written in standard English?

Reviewer #1: Yes

5. Review Comments to the Author

Reviewer #1: 1- Study setting (methodology section)

Spelling mistake in line 3

2- Explain the sampling technique in methodology section (no sampling technique)

3- What is the aim of calculating sample size for each objective separately?

4- What is meant by kebele?

5- Why you apply pilot testing on 5% of the sample?

6- Why you consider 50% as a cut off point for dividing the people into good or poor in knowledge, attitude, Social distancing practice, ------etc?

As these are means, you can consider those who have above the mean as good and below the mean as poor

7- More explanation of the data collection tool is needed regarding knowledge, attitude, Social distancing practice, ------etc?

8- Rewriting of the references is needed

9- What is meant by Merchant in the occupational status (should be mentioned)?

10- What is currency of the monthly income?

11- What is the value of asking about housing tenure, TV and Radio in sociodemographic characteristics?

12- In this study, 38.3% [95% CI:33.5%, 43.1%)] of the study participants have good social distancing practices for the prevention of COVID-19.

Where does this finding present in your results?

13- The title of the manuscript should be broadened because social distance practice and associated factors are present in table 10, 11 only.

6. PLOS authors have the option to publish the peer review history of their article (what does this mean?). If published, this will include your full peer review and any attached files.

Reviewer #1: No

---

## [Author Response · Author response to Decision Letter 0]

15 Jun 2021

9 JUNE, 2021

Dear, Prasenjit Mitra

Academic Editor

PLOS ONE

I am very delighted in your swift response to let me know that our manuscript has passed through the review stage and sent back to us so as to make a revision based on the comments given by yours well-qualified, experienced and constructive reviewers. Therefore, I have revised the manuscript according to the reviewers' comments. The revision process was conducted by a point-by-point response to each and every comments and questions mentioned by the respected reviewers. I have highlighted all the amendments in our 'Revised Manuscript with Track Changes' file. I have uploaded the comment of the editor in the cover letter section. Likewise, the response to the respected reviewers file also uploaded separately in the ‘RESPONSE TO REVIEWERS’ section. Moreover, I have made an amendment to our manuscript in line with your comments.

COMMENT-1 

• Please ensure that your manuscript meets PLOS ONE's style requirements 

ANSWER: I have checked the format of the revised manuscripts in your submission guidelines.

COMMENT-2: 

• Please include additional information regarding the survey or questionnaire used in the study and ensure that you have provided sufficient details that others could replicate the analyses

o ANSWER: I have attached the questionnaire

COMMENT-3: 

• Please provide additional details regarding participant consent….

o ANSWER: The ethics committees had approved the verbal consent procedure. 

• If your study included minors, state whether you obtained consent from parents or guardians.

o ANSWER: Our study did not include minors

• Why written consent was not obtained

o ANSWER. Most of our study participants in the research were illiterate and also asking them to review and sign the forms during such COVID-19 pandemic was considered risky. The research presents no harm to the subjects, involves no procedures and the topic by itself was not sensitive for which written consent was required. So, the response was obtained by informed verbal consent from each study participant. 

COMMENT-4: 

• Please upload a copy of Supporting Information S1 File and S2 File Which you refer to in your text on page 15.

ANSWER. I have uploaded the Supporting Information S1 File and S2 File

Sincerely,

Anteneh Fikrie (BSc, MPH)

Corresponding author

address: antenehfikrie3@gmail.com mobile: +251-922-465-129

---

## [Decision Letter · Decision Letter 1]

25 Oct 2021

PONE-D-21-13624R1Social distancing practice and associated factors in response to COVID-19 pandemic at West Guji Zone, Southern Ethiopia, 2021: A community based cross-sectional studyPLOS ONE

Dear Dr. Fikrie,

Thank you for submitting your manuscript to PLOS ONE. After careful consideration, we feel that it has merit but does not fully meet PLOS ONE’s publication criteria as it currently stands. Therefore, we invite you to submit a revised version of the manuscript that addresses the points raised during the review process.

We look forward to receiving your revised manuscript.

Kind regards,

Jianguo Wang, PhD

Academic Editor

PLOS ONE

Reviewers' comments:

Reviewer's Responses to Questions

**Comments to the Author**

1. If the authors have adequately addressed your comments raised in a previous round of review and you feel that this manuscript is now acceptable for publication, you may indicate that here to bypass the “Comments to the Author” section, enter your conflict of interest statement in the “Confidential to Editor” section, and submit your "Accept" recommendation.

Reviewer #2: (No Response)

2. Is the manuscript technically sound, and do the data support the conclusions?

Reviewer #2: Partly

3. Has the statistical analysis been performed appropriately and rigorously? 

Reviewer #2: (No Response)

4. Have the authors made all data underlying the findings in their manuscript fully available?

Reviewer #2: (No Response)

5. Is the manuscript presented in an intelligible fashion and written in standard English?

Reviewer #2: (No Response)

6. Review Comments to the Author

Reviewer #2: Thank you for carried out this study in the context of COVID-19 pandemic.

well written though some things to explain and add in the text to fit the journal requirement:

1. Introduction:

Add literature in the introduction about research done in Africa bout KAP showing the implication of social distance on limitation of COVID-19 transmission

can you please show us the importance clearly of social distance ( SD) in the covid-19 limitation of transmission without using face mask and in which percentage the SD itself can limit the disease propagation in the community?

You conclude the introduction by showing the purpose of the study but in which situation was this study carried out since the some countries in the African continent have underwent series of lockdown as a major mean to limit the disease spread, can please the authors provide the time in which the country as a justification of this study?

2.Method

Please follow guideline of presenting the methodology part as per the journal PlosOne

is it possible to talk about the place where the study was carried out in term of activities and some more detail of the region in the methodology

Please try to deeply describe the sampling technique in this study, which technique was used and how and why the sample size was not got in 100% (410 instead of 447??); describe properly and with detail the population included in the study and the place of data collection according to geographical distribution per km/population

You mentioned that the questionnaire was made in English and translate in local language, what was the power of validating this questionnaire after translation?? and then after collecting the data was the questionnaire translated back in English or not? if yes, then please mention it in the text.

The result is not well presented and the knowledge is not about social Distance but about COVID-19, can the authors please revise the result according to the main objective of the study and discuss accordingly ( demographics, KAP, Factors and bivariate and multivariate analyses; Noted that bivariate and multivariate in the same table per group vis via KAP and factors). Discussion should be presented accordingly too.

Please, provide limitations and strength of this study at the end of the discussion and please do not repeat the result in the conclusion but provide the take away message from this study and suggestions to local people, government and future researchers

Thank very much

7. PLOS authors have the option to publish the peer review history of their article (what does this mean?). If published, this will include your full peer review and any attached files.

Reviewer #2: **Yes: **Franck Katembo Sikakulya

---

## [Author Response · Author response to Decision Letter 1]

11 Nov 2021

November 12, 2021

Manuscript ID: PONE-D-21-13624

Manuscript title: Social Distancing Practice and Associated Factors in Response to COVID-19 Pandemic at West Guji Zone, Southern Ethiopia, 2020 A Community Based Cross-sectional study

Dear respected reviewers, 

Ref: A point-by-point response to the comments

Dear reviewer First of all I would like to thank you very much for your appreciation and forwarding constructive and insightful comments which we believe that it would improve the quality of our manuscript. Thus, please find in the table below the amendments and responses based on your comments, suggestions and questions. We hopefully believe that the current version of our manuscript has met the concerns of the reviewers and your editorial team.

Number Questions/comments from reviewer Answer from author Lines 

1 Introduction

• Can you please show us the importance clearly of social distance (SD) in the covid-19 limitation of transmission without using face mask and in which percentage the SD itself can limit the disease propagation in the community? I thank you really for this comment. I have incorporated the sentence accordingly.

 99-106

2 • Add literature in the introduction about research done in Africa bout KAP showing the implication of social distance on limitation of COVID-19 transmission Thank you your insightful comment. I have added some literatures according to your comments. 107-113

3 • You conclude the introduction by showing the purpose of the study but in which situation was this study carried out since the some countries in the African continent have underwent series of lockdown as a major means to limit the disease spread, can please the authors provide the time in which the country as a justification of this study? Thank you for the question. The revised draft of the manuscript has been modified. 114-121

4 Methods

• Is it possible to talk about the place where the study was carried out in term of activities and some more detail of the region in the methodology?

 Thanks very much for your concern. I have reduce the details of the region

 125-128

5 • Please try to deeply describe the sampling technique in this study, which technique was used? 

 We have employed simple random sampling technique and tried to describe in the manuscript document. 160-166

6 • How and why the sample size was not got in 100% (410 instead of 447??); 

 I have described this in the result section. The study response rate was 410 (91.7%), this mean that the remaining 37(8.3%) were not voluntarily to be participated in the study. 259-260

7 • Describe properly and with detail the population included in the study and the place of data collection according to geographical distribution per km/population

 I have described this in the methods section under Study population, sample size determination and procedure sub-section 143-146

8 • You mentioned that the questionnaire was made in English and translate in local language, what was the power of validating this questionnaire after translation?? and then after collecting the data was the questionnaire translated back in English or not? if yes, then please mention it in the text. The questionnaire was translated back in to English. I have mentioned this statement in the methods section.

 174

9 • The result is not well presented and the knowledge is not about social Distance but about COVID-19, can the authors please revise the result according to the main objective of the study and discuss accordingly (demographics, KAP, Factors and bivariate and multivariate analyses; noted that bivariate and multivariate in the same table per group vis via KAP and factors). 

 Dear respected reviewer, thank you very much for your insightful comments you raised here. I have presented the results according to your comments. 

 304-372

10 • Discussion should be presented accordingly too. Thank you again for your constructive comments. I have presented the discussion based on your comments. 376-480

11 • Please, provide limitations and strength of this study at the end of the discussion Thank you. I have incorporated the limitations and strength of the study.

 481-488

12. Please do not repeat the result in the conclusion but provide the take away message from this study and suggestions to local people, government and future researchers

 Thank you for your politeness. I have wrote the conclusion without repeating the result. 490-500

---

## [Decision Letter · Decision Letter 2]

15 Nov 2021

PONE-D-21-13624R2Social distancing practice and associated factors in response to COVID-19 pandemic at West Guji Zone, Southern Ethiopia, 2021: A community based cross-sectional studyPLOS ONE

Dear Dr. Fikrie,

Thank you for submitting your manuscript to PLOS ONE. After careful consideration, we feel that it has merit but does not fully meet PLOS ONE’s publication criteria as it currently stands. Therefore, we invite you to submit a revised version of the manuscript that addresses the points raised during the review process.

Please pay more attentions to the writing-up.

We look forward to receiving your revised manuscript.

Kind regards,

Jianguo Wang, PhD

Academic Editor

PLOS ONE

Journal Requirements:

Reviewers' comments:

Reviewer's Responses to Questions

**Comments to the Author**

1. If the authors have adequately addressed your comments raised in a previous round of review and you feel that this manuscript is now acceptable for publication, you may indicate that here to bypass the “Comments to the Author” section, enter your conflict of interest statement in the “Confidential to Editor” section, and submit your "Accept" recommendation.

Reviewer #2: All comments have been addressed

2. Is the manuscript technically sound, and do the data support the conclusions?

Reviewer #2: Yes

3. Has the statistical analysis been performed appropriately and rigorously? 

Reviewer #2: Yes

4. Have the authors made all data underlying the findings in their manuscript fully available?

Reviewer #2: Yes

5. Is the manuscript presented in an intelligible fashion and written in standard English?

Reviewer #2: Yes

6. Review Comments to the Author

Reviewer #2: Thank you for making the manuscript more better than the first draft; However, some comments :

1. In the introduction, please update the figures of COVID-19 cases (Lines 69-70)

2. Methodology: Please arrange this part as per PlosOne guideline and also remove the ethical part (lines 134-139) in the study period and population parts but put it in a separate part named "Ethical approval" and aslo the remaining part meaning lines 139-143 in data collection procedures and quality part

3. Please write the word COVID-19 in capital throughout the text

Thanks for the great work

7. PLOS authors have the option to publish the peer review history of their article (what does this mean?). If published, this will include your full peer review and any attached files.

Reviewer #2: **Yes: **Franck Katembo Sikakulya

---

## [Author Response · Author response to Decision Letter 2]

16 Nov 2021

November 15, 2021

Manuscript ID: PONE-D-21-13624R2

Manuscript title: Social Distancing Practice and Associated Factors in Response to COVID-19 Pandemic at West Guji Zone, Southern Ethiopia, 2020 A Community Based Cross-sectional study

Dear respected reviewer-2, 

It is my pleasure to bestow my gratitude for your genuine appreciation on our second revised manuscript document. Moreover, the comments you gave us yet would also improve the quality of our manuscript. Thus, please find the following amendments and responses based on your insightful and constructive comments, and suggestions. 

Comments from reviewer-2

1. In the introduction: 

1. Please update the figures of COVID-19 cases (Lines 69-70)

Answer: Really, I would like to forward a big salute for this comment. I have updated the figures according to the following. As of 1 December, more than 254 million cases and 5.1 million deaths have been reported globally until 15 November 2021.

2. Methodology:

a. Please arrange this part as per PlosOne guideline and also remove the ethical part (lines 134-139) in the study period and population parts but put it in a separate part named "Ethical approval"

i. Answer: Thank you again for your comment. I have removed the ethical consideration part and put as a separate subheading below to the data processing and analysis section. 

b. Also the remaining part meaning lines 139-143 in data collection procedures and quality part

Answer: Here also, I have moved and included the remaining sentences under the data collection procedures and quality section.

3. Please write the word COVID-19 in capital throughout the text

Answer: I have appreciated your wonderful, deep and scientific comments. I do have changed the word COVID-19 to the capital letter throughout our manuscript.

Once again I would like to say thank you very much for your astonishing, scientific and insightful comments you gave us on our manuscript.

---

## [Decision Letter · Decision Letter 3]

24 Nov 2021

Social distancing practice and associated factors in response to COVID-19 pandemic at West Guji Zone, Southern Ethiopia, 2021: A community based cross-sectional study

PONE-D-21-13624R3

Dear Dr. Fikrie,

We’re pleased to inform you that your manuscript has been judged scientifically suitable for publication and will be formally accepted for publication once it meets all outstanding technical requirements.

Kind regards,

Jianguo Wang, PhD

Academic Editor

PLOS ONE

Additional Editor Comments (optional):

Reviewers' comments:

Reviewer's Responses to Questions

**Comments to the Author**

1. If the authors have adequately addressed your comments raised in a previous round of review and you feel that this manuscript is now acceptable for publication, you may indicate that here to bypass the “Comments to the Author” section, enter your conflict of interest statement in the “Confidential to Editor” section, and submit your "Accept" recommendation.

Reviewer #2: All comments have been addressed

2. Is the manuscript technically sound, and do the data support the conclusions?

Reviewer #2: Yes

3. Has the statistical analysis been performed appropriately and rigorously? 

Reviewer #2: Yes

4. Have the authors made all data underlying the findings in their manuscript fully available?

Reviewer #2: Yes

5. Is the manuscript presented in an intelligible fashion and written in standard English?

Reviewer #2: Yes

6. Review Comments to the Author

Reviewer #2: thank you for providing the revised manuscript of this great work

however, some comments

change covid-19 line 487 to COVID-19 in limitations and strengths of the draft

7. PLOS authors have the option to publish the peer review history of their article (what does this mean?). If published, this will include your full peer review and any attached files.

Reviewer #2: **Yes: **Franck Katembo Sikakulya

---

## [Editor Report · Acceptance letter]

26 Nov 2021

PONE-D-21-13624R3 

*Social distancing practice and associated factors in response to COVID-19 pandemic at West Guji Zone, Southern Ethiopia, 2021: A community based cross-sectional study*

Dear Dr. Fikrie:

I'm pleased to inform you that your manuscript has been deemed suitable for publication in PLOS ONE. Congratulations! Your manuscript is now with our production department. 

Kind regards, 

on behalf of

Dr. Jianguo Wang 

Academic Editor

PLOS ONE